# [RE] Double-Hard Debias: Tailoring Word Embeddings for Gender Bias Mitigation

## Reproducibility Summary

### Scope of Reproducibility

The authors claim that the frequency of the words in the training corpus contributes to gender bias in the embeddings. Removing this frequency component from embeddings along with neutralizing the gender component yields gender debiased embeddings with new benchmarks on gender bias quantifying metrics.

### Methodology

We use the authors code and verify the algorithm provided in the paper for consistency. The double-hard debias algorithm is a post-training algorithm. After applying this algorithm, we test the results on the different datasets used by the authors to benchmark it. We use the free google colab to run these experiments. We add comments and rename variables to improve the readability of the code in our release [1].

### Results

The authors use two sets of evaluations to prove the efficacy of their algorithm. First, they use neighborhood metric, WEAT, and co-reference resolution task to quantify the gender bias in embeddings. We were not able to reproduce the latter task of co-reference resolution owing to the difficulty in the readability of the code. Moreover, we report that the neighborhood metric test is not reproducible with the information provided by the authors in their paper and code. We try to reproduce this by filling in our own assumptions but get drastically different results. Second, they test their word embedding quality on existing benchmarking tasks - word analogy and concept categorization. This part is reproducible to within 0.5% of the reported value.

### What was easy

The author's code readability is low, which we modify in our implementation. Other than that, the code is provided in form of notebooks that run on the latest versions of all libraries. We run these notebooks on the free google colab, making it economically feasible to reproduce. So code and results are essentially easy to re-implement.

### What was difficult

It was difficult to map the algorithm provided in the paper to the code implementation due to poor code writing standards. The neighborhood metric is difficult to implement as authors do not provide a random state which in turn is varying the results. The list of constants should be added separately to ease the running of various experiments. Moreover, we were not able to reproduce the co-reference resolution test for measuring bias in embedding. The code provided by the authors for this experiment is difficult to understand and execute.

### Communication with original authors

We did not have any communication with the original authors.

---

[1] https://anonymous.4open.science/r/74f2e710-e657-474d-a40b-e89af2790c57/

# 1   Introduction

Despite widespread use in natural language processing (NLP) tasks, word embeddings have been criticized for inheriting unintended gender bias from training corpora. [1] highlight that in word2vec embeddings trained on the Google News dataset [2], "programmer" is more closely associated with "man" and "homemaker" is more closely associated with "woman". Such gender bias has also been shown to propagate in downstream tasks. Despite plenty of work in this field, with methods ranging from corpus level modifications to post-training modifications to embeddings, it remains an unsolved problem. With this work, the authors combine two techniques to reduce gender bias in embeddings. First, they argue that the frequency of words in the corpus adds to the bias. And thus use the work of [3] to remove the frequency component from trained embeddings. Second, they use the hard debias algorithm of [1], to remove the gender direction from the trained embeddings of most biased words. Combining these two techniques, they benchmark the result of their algorithm by showcasing reduction in bias and limited loss of information in the resultant word embeddings.

# 2   Scope of reproducibility

The authors claim that the frequency of words in the training corpus contributes towards gender bias in the embeddings. Removing this frequency component from embeddings along with neutralizing the gender component yields gender debiased embeddings with new benchmarks.

- Claim 1: The double hard debias algorithm reduces gender bias significantly. This is verified on 3 benchmarking datasets described in the section 3.2 below. We showcase the outcome of our experiments of these in Table 1 and Table 2.
- Claim 2: The above post-processing algorithm of gender debiasing doesn't hamper the inherent use-case of word embeddings. This is verified on standard embedding quality measurement techniques described below. We present the results of our experiments on these in Table 3.

Each subsection in section 4 refers to above claims and talks about the level and ease of reproducibility of above claims and experiments as performed by the authors for these claims.

# 3   Methodology

We use the authors code to ease our understanding of the experiments and to reproduce the claims presented by the author. We used google colab for re-running these experiments. For complete understanding of the algorithm, we used the mixture of paper and code.

## 3.1   Model descriptions

The authors introduce the double hard debias algorithm in this paper. This is a post-training algorithm that works after the embeddings have been trained to reduce the gender bias in those embeddings. Hence, this algorithm requires no parameters to train. First, the frequency information from these embeddings is removed. This is done by calculating the first $k$ principal components of the trained embeddings. The value of $k$ is empirically determined. These projections of embeddings along these $k$ components are then removed from the embeddings. Second, the gender direction is determined by averaging the difference of 10 gender pair words. Then the projection of embeddings along this gender direction is removed. The double hard debias is now done.

## 3.2   Datasets and Experimental Setup

The authors perform two sets of experiments to highlight the efficacy of their approach. In the first set, they prove the reduction in gender bias through 3 methods: co-reference resolution via the [4] and the OntoNotes 5.0 dataset, the WEAT, the NeighbourHood Metric.

- **Co-reference Resolution**: Coreference resolution aims at identifying noun phrases referring to the same entity. [4] identified gender bias in modern coreference systems, e.g. "doctor" is prone to be linked to "he" and also created a new WINO bias dataset to quantify the bias in word embeddings.
- **WEAT**: The Word Embeddings Association Test is a permutation test used to measure bias. The authors consider male names and females names as attribute sets and compute the differential association of two sets of target words as used in [5] and the gender attribute sets.

| Embeddings | Career & Family | | Math & Arts | | Science & Arts | |
|---|---|---|---|---|---|---|
| | $d$ | $p$ | $d$ | $p$ | $d$ | $p$ |
| GloVe | 1.81 | 0.0 | 0.55 | 0.14 | 0.88 | 0.04 |
| GN-GloVe | 1.82 | 0.0 | 1.21 | $6e^{-3}$ | 1.02 | 0.02 |
| GN-GloVe$(w_a)$ | 1.76 | 0.0 | 1.43 | $1e^{-3}$ | 1.02 | 0.02 |
| GP-GloVe | 1.81 | 0.0 | 0.87 | 0.04 | 0.91 | 0.03 |
| GP-GN-GloVe | 1.80 | 0.0 | 1.42 | $1e^{-3}$ | 1.04 | 0.01 |
| Hard-GloVe | 1.55 | $2e^{-4}$ | 0.07 | 0.44 | 0.16 | 0.62 |
| Strong Hard-GloVe | 1.55 | $2e^{-4}$ | 0.07 | 0.44 | 0.16 | 0.62 |
| Double-Hard GloVe | 1.53 | $2e^{-4}$ | 0.09 | 0.57 | 0.15 | 0.61 |

Table 1: WEAT test of embeddings before/after Debiasing. The bias is insignificant when p-value, $p > 0.05$. Lower effective size ($d$) indicates less gender bias. Significant gender bias related to Career & Family and Science & Arts words is effectively reduced by Double-Hard GloVe. Note for Math & Arts words, gender bias is insignificant in original GloVe.

- **Neighbourhood Metric**: Introduced by [6], this is a metric to measure bias by clustering. The authors take the top k most biased words according to their cosine similarity with gender direction in the original GloVe [7] embedding space. They then run k-Means to cluster them into two clusters and compute the alignment accuracy with respect to gender, results are presented in Table 2. The lower the accuracy, the less the gender bias in the embeddings.

In the second set, the authors prove the information retention of the embeddings post applying their algorithm. They use two tasks for it: word analogy task and concept categorization task.

- **Word Analogy**: Given three words A, B and C, the analogy task is to find word D such that "A is to B as C is to D". In the experiments, D is the word that maximize the cosine similarity between D and C - A + B. The authors evaluate all non-debiased and debiased embeddings on the MSR [8] word analogy task, which contains 8000 syntactic questions, and on a second Google word analogy [9] dataset that contains 19,544 (Total) questions, including 8,869 semantic (Sem) and 10, 675 syntactic (Syn) questions.

- **Concept Categorization**: The goal of concept categorization is to cluster a set of words into different categorical subsets. For example, "sandwich" and "hotdog" are both food and "dog" and "cat" are animals. The clustering performance is evaluated in terms of purity [10] - the fraction of the total number of the words that are correctly classified. Experiments are conducted on four benchmark datasets: the Almuhareb-Poesio (AP) dataset [11]; the ESSLLI 2008 [12]; the Battig 1969 set [13] and the BLESS dataset [14].

All of the above are standard datasets and evaluation methods which do not require any post-processing and can be directly used for testing any word embedding. Our code used to replicate the above experiments can be found here.[2]

### 3.3 Computational requirements

We used the free google colab to run our experiments. Apart from the data download time, all these experiments run within 30 mins on the free google GPU setup. For experimenting with various variants of Glove Embedding, we use the link[3] provided by the authors.

## 4 Results

Barring the two tests in claim 1 that highlight the reduction in gender bias of their method, we were able to reproduce all other results published by the authors and thus were able to fully verify claim 2.

---

[2]https://anonymous.4open.science/r/74f2e710-e657-474d-a40b-e89af2790c57/

[3]http://www.cs.virginia.edu/ tw8cb/word_embeddings/

| Embeddings | Top 100 | | Top 500 | | Top 1000 | |
|---|---|---|---|---|---|---|
| | Ours | Authors | Ours | Authors | Ours | Authors |
| GloVe | 100.0 | 100.0 | 100.0 | 100.0 | 100.0 | 100.0 |
| GN-GloVe | 100.0 | 100.0 | 100.0 | 100.0 | 99.8 | 99.9 |
| GN-GloVe($w_a$) | 100.0 | 100.0 | 99.5 | 99.7 | 89.4 | 88.5 |
| GP-GloVe | 100.0 | 100.0 | 100.0 | 100.0 | 100.0 | 100.0 |
| GP-GN-GloVe | 100.0 | 100.0 | 100.0 | 100.0 | 100.0 | 99.4 |
| (Strong) Hard-GloVe | 76.5 | 59.0 | 80.2 | 62.1 | 80.2 | 68.1 |
| Double-Hard GloVe | 66.5 | 51.5 | 74.1 | 55.5 | 70.4 | 59.5 |

Table 2: Clustering Accuracy (%) of top 100/500/1000 male and female words. Lower accuracy means less gender cues can be captured. Double-Hard GloVe consistently achieves the lowest accuracy.

## 4.1 Results reproducing original paper

### 4.1.1 Result 1

This section verifies the claim 1 of the authors that highlights the reduction of gender bias on 3 metrics. We successfully executed the WEAT test and got results as presented in Table 1 comparable to the ones published by authors. We were not able to reproduce 2 of these. The Neighbourhood Metric calculation is largely not reproducible because of two reasons:

1. Authors do not state whether they have normalized word vectors or not before performing this experiment.

2. Authors do not provide the random state with which they have initialised the K-means clustering which lead to different results.

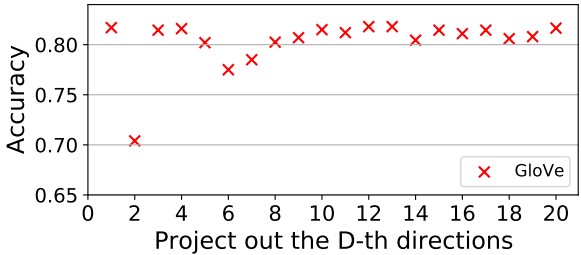

Figure 1: Clustering accuracy after projecting out D-th dominating direction and applying Hard Debias. Lower accuracy indicates less bias.

We try to replicate this using our own set of assumptions but are not able to reproduce the authors claims. We replicate it via following assumptions:

1. We experiment with both normalized and unnormalized vectors, and report the best result that came with unnormalized vectors in Table 2.

2. We experiment with various random states and report the one with best outcome.

3. We remove frequency feature along the second principal component as this is the one reported by authors in Figure 1 to have the best performance. Also, there is an unexplained mismatch between the above figure and results posted in Table 2. The best score in the above figure is close to 0.7 which is calculated on Top 1000 male and female words, but in the table, authors showcase the best result to be close to 0.59. This mismatch of outcomes is unexplained in the paper or the code.

We add the t-SNE [15] visualization comparison between the ones published by the authors and the ones which we got in Figure 5. We are unable to reproduce these visualizations as one owing to the challenges and differences in assumptions posted above.

The second result which we were not able to reproduce is the co-reference resolution task. Due to bad readability of the authors code, we were unable to execute this experiment.

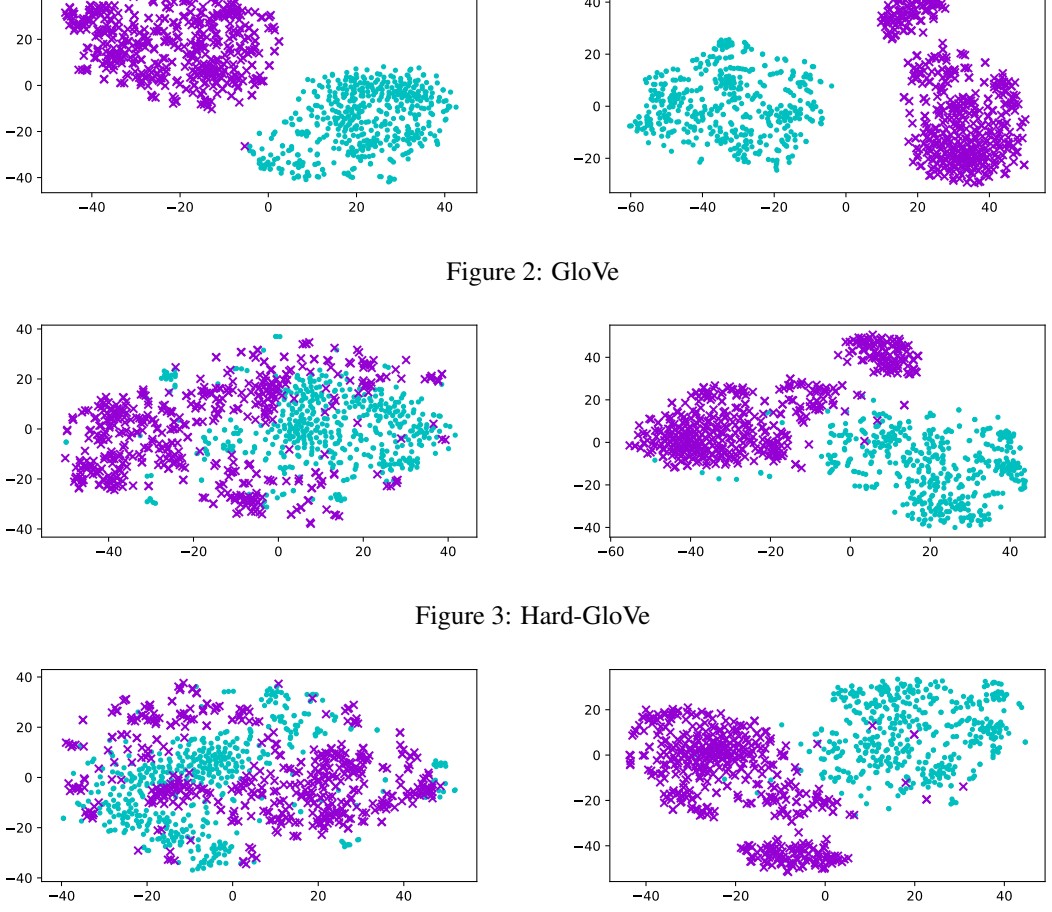

Figure 2: GloVe

Figure 3: Hard-GloVe

Figure 4: Double-Hard GloVe

Figure 5: tSNE visualization of top 500 most male and female embeddings. On the left is the authors published visualisations and on the right is what we got after during our experiments. In the Double-Hard GloVe figures, the authors showcase mixing up of the two clusters showcasing less gender bias, which does not match with our reproduction of the same experiment.

### 4.1.2 Result 2

This verifies claim 2 of the authors that the resultant word embeddings retain the semantic and associative information which makes this distributed word embeddings useful for natural language processing tasks. The authors use the Word Analogy task and Concept Categorization task as explained above in 3.2. We were able to reproduce the results published by authors to within 0.5% accuracy and present the outcomes in Table 3.

### 4.2 Results beyond the paper

In here, we present the qualitative analysis we did to measure the gender bias aspect of the word embeddings. We draw comparison with heavily biased words and their association with gender pair words - he and she. In Table 4, we present the difference in cosine similarity of a few biased words with respect to 'he' and 'she'. With this we try to showcase that the authors' algorithm has indeed contribute towards reduced gender bias.

| Embeddings | Analogy | | | | Concept Categorization | | | |
|---|---|---|---|---|---|---|---|---|
| | Sem | Syn | Total | MSR | AP | ESSLI | Battig | BLESS |
| GloVe | 80.5 | 62.8 | 70.8 | 54.2 | 56.1 | 72.7 | 50.0 | 81.0 |
| GN-GloVe | 77.6 | 61.6 | 68.9 | 51.8 | 56.9 | 75.0 | 47.6 | 85.0 |
| GN-GloVe($w_a$) | 77.7 | 61.6 | 68.9 | 51.9 | 56.9 | 72.7 | 50.2 | 82.5 |
| GP-GloVe | 80.6 | 61.7 | 70.3 | 51.3 | 56.1 | 72.7 | 49.0 | 78.5 |
| GP-GN-GloVe | 77.6 | 61.7 | 68.9 | 51.8 | 61.1 | 70.4 | 50.9 | 77.5 |
| Hard-GloVe | 80.3 | 62.7 | 70.7 | 54.3 | 62.3 | 79.5 | 48.2 | 84.5 |
| Strong Hard-GloVe | 78.9 | 62.4 | 69.8 | 53.9 | 62.3 | 79.5 | 50.9 | 84.5 |
| Double-Hard GloVe | 80.9 | 61.6 | 70.4 | 53.8 | 59.6 | 72.7 | 46.7 | 79.5 |

Table 3: Results of word embeddings on word analogy and concept categorization benchmark datasets. Performance (x100) is measured in accuracy and purity, respectively. On both tasks, there is no significant degradation of performance due to applying the proposed method.

| Word | Before | After |
|---|---|---|
| *doctor* | 0.013 | 0.01 |
| *programmer* | 0.036 | −0.007 |
| *homemaker* | −0.112 | 0.033 |
| *nurse* | −0.121 | 0.033 |
| *worker* | −0.007 | 0.023 |
| *president* | 0.083 | 0.034 |
| *politician* | 0.066 | 0.029 |

Table 4: Qualitative Analysis for some highly biased words before and after using the double hard debiasing. Negative means that the words are biased towards 'she' and positive means that the words are biased 'he'.

# 5 Discussion

The authors present a viable post-training method to reduce gender bias from non-contextual word embeddings. The author uses 3 benchmarks to showcase a reduction in gender bias. However, we were only able to reproduce only 1 of the benchmarks, with different results on the neighborhood metric.

We were strongly able to reproduce the experiments that validate claim 2 of the paper, which showcases that the paper's double debias algorithm doesn't hamper the useful properties of word embeddings.

## 5.1 What was easy

The authors code for claim 2 and double debias algorithm was easy to run as it was shared in the form of jupyter notebook. The pseudo-code for the algorithm was easy to understand and this made it easier to follow in the give code. The authors structured the claims in the paper very well, which made it easier to match experiments with these claims.

## 5.2 What was difficult

The authors code lacked structure for claim 1 and other sub parts of the paper, and thus it was difficult to follow. For the co-reference resolution task, a sub part of claim 1, we spent a lot of time to execute the reference code but we were still unable to execute the experiment owing to the poor code organization and readability.

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
