# OpenReview forum: "[RE] Double-Hard Debias: Tailoring Word Embeddings for Gender Bias Mitigation"
_ML_Reproducibility_Challenge/2020 — Reject_

### Official Review · AnonReviewer3 · 2021-02-25
**Review of Double-Hard Debias: Tailoring Word Embeddings for Gender Bias Mitigation**

**Rating:** 7
**Confidence:** 2

**Review:**

- Reproducibility Summary: the paper includes the reproducibility summary on the first page. The summary is clear, well written, and major findings (i.e. failure in reproducing part of the paper, but successful replication on the rest) are incorporated in the summary.

- Scope of Reproducibility: it is well described, specifically the authors investigate an approach to generate debiased embeddings by removing the frequency component of word embeddings. The authors list 2 reproduced/verified claims from the original paper.

- Code: the authors tried to re-use the code provided by the original authors. Unfortunately, the code was not well documented and the authors struggled in re-running it. This hampered the reproducibility effort. The code provided with this paper is a revised version of the original code, complemented with more detailed comments and documentation.

- Communication with original authors: the authors did not have any communication with the original authors.

- Hyperparameter Search: the authors do not perform any hyperparameter search, since the reproduced algorithm is a post-training algorithm which does not require to train any parameter.

- Ablation Study: there is no ablation study, but it does not make sense to have it.

- Discussion on results: the experimental results are well presented and compared with the original results. The authors describe easy parts and challenges in reproducing the original paper. Due to difficulties in interpreting the original code the authors could not reproduce part of the experiments.

- Recommendations for reproducibility: the authors do not mention any recommendations for the original authors, but they explicitly list missing details that were important for the reproducibility.

- Results beyond the paper: the authors present some results beyond the original paper: they report a qualitative analysis with some biased words comparing the results before and after debiasing.

- Overall organization and clarity: the paper is clear and well organized. I list some typos in the following:
    - Line 27: weren’t -> were not
    - Line 33: [1] highlight -> use \citet
    - Line 49: doesn’t -> does not
    - Line 71: [4] identified -> use \citet
    - Line 82: tasks for it : -> extra blank space
    - Line 82: cateogorization -> categorization
    - Line 83: Word Analogy : -> extra space
    - Line 88: Concept Categorization : -> extra space
    - Line 114: set of assumption -> set of assumptions
    - Line 115: assumptions : -> extra space
    - Caption of Figure 5: doesn’t -> does not
    - Line 137: the difference is cosine similarity -> something is wrong in this sentence
    - Line 144: doesn’t -> does not


Overall evaluation:
Pros:
- The paper is clear and well written;
- The authors could reproduce some of the results of the original paper;
- The authors provide an elaborate list of details that are missing from the original paper.

Cons:
- The authors could not reproduce part of the experiments (co-reference resolution task) due to the poor quality of the code provided, however they could try to re-implement part of the original approach.


**Familiar With The Original Paper:**

I have not read the original paper

**Reproducibility Summary:**

Report has summary

---

### Official Review · AnonReviewer2 · 2021-02-28
**Reproducibility Report for Double-Hard Debias: Tailoring Word Embeddings for Gender Bias Mitigation**

**Rating:** 9
**Confidence:** 4

**Review:**

The authors attempted to reproduce the claims in the paper "Double-Hard Debias: Tailoring Word Embeddings for
Gender Bias Mitigation" by Wang et al. 2020

+ the reproducibility report is clearly organized.
+ it was clear in the paper how the two main claims of the original article were reproduced.
+ it was also clear how the reproducibility experiments were carried out and that the authors did not introduce any experimenter or measurement bias themselves in the process.
+ it was evident which claim was easier to reproduce and why and what was difficult.

- it was not clear whether the authors attempted to reach out to the original authors to validate some of their assumptions
- likewise, they mentioned difficulty in understanding/executing code but it was not clear whether the authors were contacted.

**Familiar With The Original Paper:**

I have read the original paper

**Reproducibility Summary:**

Report has summary

---

### Official Review · AnonReviewer1 · 2021-03-02
**good, but lacks substance**

**Rating:** 5
**Confidence:** 4

**Review:**

Authors worked on replicating the paper by Wang et al. (2020) on tailoring embeddings for gender bias mitigation. Efforts were put to replicate the neighborhood metric test, however, the results were not reproducible due to lack of clarity in information/code by the original authors. Analysis on word embedding quality was reproducible within 5% of the original reported value, however, authors were not able to attempt replicating the coreference resolution experiments due to unreadable codes as they claimed. No communication was attempted with the original authors to clarify the codebase for replication purposes. Overall, while the carried out experiments were solid (additional qualitative analysis on gender bias aspect was helpful) and the shared codebase for the replicated part would be useful to the researchers, I thought the work lacked substantial materials to be beneficial to the community based on its current standings.

**Familiar With The Original Paper:**

I have not read the original paper

**Reproducibility Summary:**

Report has summary

---

### Decision · Program_Chairs · 2021-03-31

**Decision:**

Reject

**Comment:**

Overall reviews and/or the paper content not good enough for the AC to recommend to the journal.